# Association of sleep duration at age 50, 60, and 70 years with risk of multimorbidity in the UK: 25-year follow-up of the Whitehall II cohort study

**Séverine Sabia** [1,2] *, **Aline Dugravot** [1], **Damien Léger** [3,4], **Céline Ben Hassen** [1], **Mika Kivimaki** [2,5], **Archana Singh-Manoux** [1,2]

**1** Université Paris Cité, Inserm U1153, Epidemiology of Ageing and Neurodegenerative diseases, Paris, France, **2** Department of Epidemiology and Public Health, University College London, London, United Kingdom, **3** Université Paris Cité, EA 7330 VIFASOM (Vigilance Fatigue Sommeil et Santé Publique), Paris, France, **4** APHP, Hôtel-Dieu, Consultation de pathologie professionnelle Sommeil Vigilance et Travail, Centre du Sommeil et de la Vigilance, Paris, France, **5** Clinicum, University of Helsinki, Helsinki, Finland

* severine.sabia@inserm.fr

**Data Availability Statement:** Data cannot be made publicly available because of ethics and IRB restrictions. However, the data are available to

## Abstract

### Background

Sleep duration has been shown to be associated with individual chronic diseases but its association with multimorbidity, common in older adults, remains poorly understood. We examined whether sleep duration is associated with incidence of a first chronic disease, subsequent multimorbidity and mortality using data spanning 25 years.

### Methods and findings

Data were drawn from the prospective Whitehall II cohort study, established in 1985 on 10,308 persons employed in the London offices of the British civil service. Self-reported sleep duration was measured 6 times between 1985 and 2016, and data on sleep duration was extracted at age 50 (mean age (standard deviation) = 50.6 (2.6)), 60 (60.3 (2.2)), and 70 (69.2 (1.9)). Incidence of multimorbidity was defined as having 2 or more of 13 chronic diseases, follow-up up to March 2019. Cox regression, separate analyses at each age, was used to examine associations of sleep duration at age 50, 60, and 70 with incident multimorbidity. Multistate models were used to examine the association of sleep duration at age 50 with onset of a first chronic disease, progression to incident multimorbidity, and death. Analyses were adjusted for sociodemographic, behavioral, and health-related factors.

A total of 7,864 (32.5% women) participants free of multimorbidity had data on sleep duration at age 50; 544 (6.9%) reported sleeping ≤5 hours, 2,562 (32.6%) 6 hours, 3,589 (45.6%) 7 hours, 1,092 (13.9%) 8 hours, and 77 (1.0%) ≥9 hours. Compared to 7-hour sleep, sleep duration ≤5 hours was associated with higher multimorbidity risk (hazard ratio: 1.30, 95% confidence interval = 1.12 to 1.50; p < 0.001). This was also the case for short sleep duration at age 60 (1.32, 1.13 to 1.55; p < 0.001) and 70 (1.40, 1.16 to 1.68; p < 0.001). Sleep duration ≥9 hours at age 60 (1.54, 1.15 to 2.06; p = 0.003) and 70 (1.51, 1.10

bona fine researchers at small fee - to cover data management costs - via a data sharing portal allowing access to undertake analyses within a secure portal, https://portal.dementiasplatform.uk/. For general data sharing enquiries, please contact whitehall2@ucl.ac.uk.

**Funding:** This project is part of the National Institute on Aging (R01AG056477 to ASM and MK, https://www.nia.nih.gov/). The Whitehall II study has been supported by grants from the National Institute on Aging, NIH (R01AG056477, RF1AG062553, to ASM and MK, https://www.nia.nih.gov/); UK Medical Research Council (R024227, S011676, K013351, to MK, https://www.ukri.org/councils/mrc/); the British Heart Foundation (RG/16/11/32334, https://www.bhf.org.uk/); the Wellcome Trust (221854/Z/20/Z, to MK, https://wellcome.org/). SS is supported by the French National Research Agency (ANR-19-CE36-0004-01, https://anr.fr/en/). The funders had no role in study design, data collection and analysis, decision to publish, or preparation of the manuscript.

**Competing interests:** The authors have declared that no competing interests exist.

**Abbreviations:** BMI, body mass index; CVD, cardiovascular disease; HES, Hospital Episode Statistics; HR, hazard ratio; NHS, National Health Service; SD, standard deviation.

to 2.08; $p = 0.01$) but not 50 (1.39, 0.98 to 1.96; $p = 0.07$) was associated with incident multimorbidity. Among 7,217 participants free of chronic disease at age 50 (mean follow-up = 25.2 years), 4,446 developed a first chronic disease, 2,297 progressed to multimorbidity, and 787 subsequently died. Compared to 7-hour sleep, sleeping ≤5 hours at age 50 was associated with an increased risk of a first chronic disease (1.20, 1.06 to 1.35; $p = 0.003$) and, among those who developed a first disease, with subsequent multimorbidity (1.21, 1.03 to 1.42; $p = 0.02$). Sleep duration ≥9 hours was not associated with these transitions. No association was found between sleep duration and mortality among those with existing chronic diseases. The study limitations include the small number of cases in the long sleep category, not allowing conclusions to be drawn for this category, the self-reported nature of sleep data, the potential for reverse causality that could arise from undiagnosed conditions at sleep measures, and the small proportion of non-white participants, limiting generalization of findings.

## Conclusions

In this study, we observed short sleep duration to be associated with risk of chronic disease and subsequent multimorbidity but not with progression to death. There was no robust evidence of an increased risk of chronic disease among those with long sleep duration at age 50. Our findings suggest an association between short sleep duration and multimorbidity.

## Author summary

### Why was this study done?

- The prevalence of multimorbidity is on the rise as reflected in over half of older adults having at least 2 chronic diseases in high-income countries, making multimorbidity a major challenge for public health.

- Both short and long sleep duration has been shown to be associated with individual chronic diseases, but their associations with multimorbidity and subsequent mortality risk remain unclear.

### What did the researchers do and find?

- We used data on more than 7,000 men and women from the Whitehall II cohort study to extract sleep duration at age 50, 60, and 70 and examined its association with incident multimorbidity over 25 years of follow-up. Role of sleep duration at age 50 in transitions from a healthy state to a first chronic disease, multimorbidity, and mortality was also examined using a multistate model.

- We found a robust association of sleep duration ≤5 hours at age 50, 60, and 70 (separate analyses) with higher risk of incident multimorbidity, while the association with sleep duration ≥9 hours was observed only when measured at age 60 and 70.

- Analysis of transitions in health states showed short sleep duration at age 50 to be associated with 20% increased risk of a first chronic disease, and with a similar increased risk of subsequent multimorbidity, but within this framework there was no clear evidence of associations with mortality.

- There was no robust association between sleep duration ≥9 hours at age 50 and risk of 1 chronic disease or multimorbidity. However, in those with a chronic condition there was some evidence of higher risk of multimorbidity.

**What do these findings mean?**

- Our comprehensive analyses of the association of sleep duration with multimorbidity and the natural course of chronic disease show short sleep duration to be associated with the onset of chronic disease and multimorbidity but not with subsequent mortality in those with chronic disease(s).

- There was no clear evidence for an association between long sleep duration at age 50 and risk of chronic disease. Rather the increased risk of multimorbidity associated with long sleep duration at older ages and in those with existing disease might reflect the need for longer sleep in those with underlying chronic conditions.

## Introduction

Approximately one third of human life is devoted to sleep, emphasizing the vital role of sleep in several physiological functions essential for health. There is also consistent evidence of an association of sleep duration with chronic diseases, such as cardiovascular disease (CVD) and cancer [1,2], and with mortality [2–4], although there remain a number of outstanding questions regarding the nature of this association. First, multiple chronic conditions often coexist within the same individual, a condition known as multimorbidity [5–8], but the association of sleep duration with multimorbidity remains poorly understood due to paucity of research and cross-sectional nature of existing studies [9–13]. It is unclear how sleep duration affects trajectories from a healthy state, to 1 or more chronic diseases, and subsequent mortality. Second, current guidelines recommend 7 to 8 hours of sleep for older adults [14] but whether both short and long sleep duration carry risk for multimorbidity remains unclear. Several biological mechanisms have been proposed to explain the role of short sleep duration in disease onset [15,16] but the role of long sleep is less well understood [3,17]. The observed risk of chronic conditions among long sleepers could be due to preexisting health conditions [15,18] or, alternatively, reflect non-restorative sleep that then affects risk of subsequent disease [15,19]. Third, as people get older, their sleep habits and sleep structure change [20]; whether sleep duration in mid and later life differentially affects subsequent risk of multimorbidity has not been investigated.

The first objective of the present study was to examine the association between sleep duration at 50, 60, and 70 years of age and incident multimorbidity, using repeat data on sleep duration and continuous assessment of chronic diseases spanning over 25 years. A second objective was to determine whether sleep duration at age 50 shapes the natural course of

chronic disease, from a healthy state, to a first chronic disease, multimorbidity, and death using multistate models to examine the association of sleep duration at age 50 with transitions between each of these health states. In these analyses, the focus is on sleep duration at age 50 as chronic conditions are less prevalent and reverse causation bias that could arise from underlying conditions affecting sleep duration is less likely. In additional analyses, we examined the association of sleep duration with the onset of multimorbidity and death in the subgroup of participants with 1 chronic condition to examine whether sleep pattern after onset of chronic conditions is associated with adverse health outcomes [15]. Finally, in post hoc analysis, the association between sleep disturbance at age 60 and 70, where we had data on these measures, and risk of incident multimorbidity was examined.

## Methods

This study is reported following the Strengthening the Reporting of Observational Studies in Epidemiology (STROBE) guideline (S1 STROBE Checklist).

### Study population

The Whitehall II study is an ongoing cohort study established in 1985 among 10,308 British civil servants (6,895 men and 3,413 women, aged 35 to 55 years) [21]. Since baseline, follow-up clinical examinations have taken place approximately every 4 to 5 years, each wave taking 2 years to complete, with the last completed wave conducted in 2015 to 2016. Except for 10 individuals, all participants (99.9%) are linked to UK National Health Service (NHS) electronic health records. The NHS provides most of the health care in the United Kingdom, including in- and out-patient care, and record linkage is undertaken using a unique NHS identifier held by all UK residents. Data from linked records were updated on an annual basis, until March 31, 2019. Written informed consent from participants and research ethics approvals were renewed at each contact; the most recent approval was from the University College London Hospital Committee on the Ethics of Human Research, reference number 85/0938.

### Sleep duration

Sleep duration was measured at 6 data collection waves, 1985 to 1988, 1997 to 1999, 2002 to 2004, 2007 to 2009, 2012 to 2013, and 2015 to 2016 using the question "How many hours of sleep do you have on an average week-night?" Response categories were: ≤5 hours, 6 hours, 7 hours, 8 hours, and ≥9 hours. For each participant, sleep duration at age 50, 60, and 70 was extracted across the data waves using data from the wave at which the participant's age was the closest to the target age, allowing a ±5 year margin for each age of interest.

Trajectories of change in sleep duration between age 50 and 70 [22] among those with at least 2 out of 3 measures of sleep duration at age 50, 60, and 70 were defined using group-based trajectory modeling using the *traj*-command in Stata [23]. Groups were chosen using the best model fit (Bayesian Information Criterion values and average posterior probabilities) and meaningful interpretation of trajectories [24].

At the 2012 wave, when participants were 60 to 83 years, an accelerometer sub-study—a one-off addition to the main data collection—was undertaken on participants who attended the central London research clinic or were assessed at home if they resided in the South-Eastern regions of England. Wrist-worn accelerometers, the GENEActiv (Activinsights, Kimbolton, UK), were worn 24 hours over 9 consecutive days [25]. Sleep duration was estimated using a validated algorithm guided by a sleep log [26]; data from the first and last nights were removed leading to data over 7 nights. Usual daily sleep duration was estimated as the mean of sleep duration over 7 nights and for those with less than 7 nights of measurement, weighted

average of sleep duration was calculated as: 5 × week night sleep duration + 2 × weekend night sleep duration)/7.

In post hoc analysis, we used data on sleep quality measured using the Jenkins sleep problems scale [27]. This measure was introduced to the study questionnaire in 1997 and repeated at following study waves, allowing us to extract data on sleep quality at age 60 and 70, but not age 50. Participants were asked how often in the past month they had experienced the following symptoms: (1) trouble falling asleep; (2) waking up several times per night; (3) trouble staying asleep (including waking far too early); and (4) disturbed or restless sleep. The following response categories were available: Not at all (scored 0), 1 to 3 days (scored 1), 4 to 7 days (scored 2), 8 to 14 days (scored 3), 15 to 21 days (scored 4), and 22 to 31 days (scored 5). The sum of these items was then used as a continuous scale to measure sleep problems. The score was further dichotomized to reflect low sleep disturbance (0 to 11) and high sleep disturbance (12 to 20) [28].

## Multimorbidity

Multimorbidity was defined as the presence of 2 or more chronic diseases out of a predefined list of 13 chronic diseases that were chosen because they are prevalent across the adult life-course. Inclusion of at least 12 conditions is thought to accurately reflect multimorbidity [29] and our list was chosen from previous research on multimorbidity [8,30]. As in previous studies, risk factors such as hypertension and obesity were not included in the list [31,32]. We identified chronic diseases using data from the Whitehall clinical examinations and via linkage to electronic health records up to March 31, 2019 from the Hospital Episode Statistics (HES) database, the Mental Health Services Data Set (which in addition to in- and out-patient data also include records on care in the community), and the national cancer registry. The chronic diseases considered [33] were:

1. diabetes (ICD10: E10-E14, reported doctor-diagnosed diabetes, use of diabetes medication, or fasting glucose ≥ 7.0 mmol/l),

2. cancer (malignant neoplasms ICD10: C00-C97),

3. coronary heart disease (ICD10: I20-I25, 12-lead resting ECG recording) [34],

4. stroke (ICD10: I60-I64, MONICA-Ausburg stroke questionnaire) [34],

5. heart failure (ICD10: I50),

6. chronic obstructive pulmonary disease (ICD10: J41-J44),

7. chronic kidney disease (ICD10: N18),

8. liver disease (ICD10: K70-K74),

9. depression (ICD10: F32, F33, or use of antidepressants),

10. dementia (ICD10: F00-F03, F05.1, G30, G31) [35],

11. mental disorders, other than depression and dementia (ICD10: F06, F07, F09, F20-48 (excluding F32: depressive episode and F33: major depressive disorder, recurrent) and F60-69 (excluding F65: paraphilias and F66: other sexual disorders)) [36],

12. Parkinson's disease (ICD10: G20), and

13. arthritis/rheumatoid arthritis (ICD10: M15-M19, M05, M06).

## Mortality

Mortality was ascertained from linked records from the British national mortality register (National Health Services Central Registry) with follow-up until March 31, 2019.

## Covariates

Covariates included sociodemographic, behavioral, and health-related factors. Sex, ethnicity, and education were drawn from the baseline examination in 1985 to 1988. Other covariates, available at each wave of data collection, were extracted concurrently to the measure of sleep duration at age 50, 60, and 70.

Sociodemographic factors included age, sex, ethnicity (response to a question using the categories "White," "South Asian," "Black," and "Other" and categorized in the analysis as White and non-White, due to the small numbers in the latter group), education (primary school or less, lower secondary school, higher secondary school, university, higher degree; treated as continuous variable), occupational position (high, intermediate, and low, representing income and status at work), and marital status (married or cohabiting, other).

Health behaviors included cigarette smoking (never smoker, ex-smoker, current smoker), alcohol consumption in the previous week (none, 1 to 14 units per week, >14 units per week), time spent in moderate and vigorous physical activity (hours per week), and frequency of fruit and vegetable consumption (less than daily, once a day, twice or more a day).

Health-related factors included hypertension (systolic ≥140 or diastolic ≥90 mmHg or use of antihypertensive medication), body mass index (BMI, categorized as <18.5, 18.5 to 24.9, 25 to 29.9, and ≥30 kg/m$^2$) calculated using height and weight measured at the clinical examination using standard clinical protocols, use of sleep medication, and prevalence of 1 of the 13 conditions considered in the definition of multimorbidity.

## Statistical analysis

The analysis plan was developed prior to data analysis (S1 Text). The analyses referred to as post hoc analyses were in response to suggestions from reviewers.

## Association between sleep duration at different ages and incident multimorbidity

The association of sleep duration at age 50, 60, and 70 with incident multimorbidity was examined in separated models. The analyses were undertaken using Cox proportional-hazards regression with age as the timescale in participants free from multimorbidity at the measurement of sleep duration. Data were censored at date of multimorbidity diagnosis, death to account for competing risk [37], or March 31, 2019, whichever came first. In analysis of sleep duration at age 50, age at the beginning of the follow-up was the age at clinical assessment closest to 50 years from which the sleep duration measure and covariates were drawn. A similar approach was used in analyses of sleep duration at age 60 and 70. The proportional hazards assumption was verified using Schoenfeld residuals. Analyses were first unadjusted (age as timescale; Model 1), then adjusted for sociodemographic measures (Model 2), and finally additionally for behavioral and health-related factors (Model 3).

To examine the robustness of our findings, we repeated the main analysis (1) in participants free from any of the 13 chronic diseases used to define multimorbidity, (2) excluding users of sleep medication and (3) examined the association between accelerometer-assessed sleep duration at mean age 69 (range = 60 to 83) years and incident multimorbidity. The covariates were drawn from the 2012 wave of data collection, concurrent to the accelerometer measure. Given

the detailed data on sleep duration extracted from the accelerometer, we used restricted cubic spline regressions with Harrell knots [38], Stata command *partpred* [39], with 7-hour sleep as the reference to assess the shape of the association between sleep duration and multimorbidity risk.

Several post hoc analyses were conducted. First, the main analyses were repeated using inverse probability weighting to account for missing data. Second, we explored whether findings were driven by one specific chronic disease by repeating the analysis on sleep duration at age 50, 60, and 70 and incident multimorbidity, excluding one chronic disease at a time from the definition of multimorbidity. Third, we examined the association of trajectories of sleep duration between age 50 and 70 with incident multimorbidity with age of entry and covariates drawn from the wave sleep measure at age 70 was extracted. Fourth, the association of sleep disturbance at age 60 and age 70 with incident multimorbidity was investigated.

## Association of sleep duration with transitions to multimorbidity and death

Among participants at age 50, free from the 13 chronic diseases considered here, multistate models were used (Fig 1) to determine the association of sleep duration at age 50 with transitions from: (1) a healthy state to a first chronic disease (any from the list of 13 diseases considered); (2) a healthy state to death (in those who remained free from any of the 13 diseases during follow-up); (3) a first chronic disease to multimorbidity; (4) a first chronic disease to death; and (5) multimorbidity to death, with follow-up starting at age 50. The advantage of multistate models is that they take into account the time spent within each health state to estimate probabilities of transitions between each state. For comparison, we also examined (post hoc analysis) the association between sleep duration at age 50 and risk of mortality in the same study sample (participants free from chronic disease at age 50) irrespective of incidence of chronic disease over the follow-up.

In additional analyses, we examined the association of sleep duration after onset of a first chronic disease with transitions to multimorbidity and death, again using a multistate model. The follow-up here started at the measure of first sleep duration following the onset of a first chronic disease. In sensitivity analysis, we used inverse probability weighting to account for missing data [22].

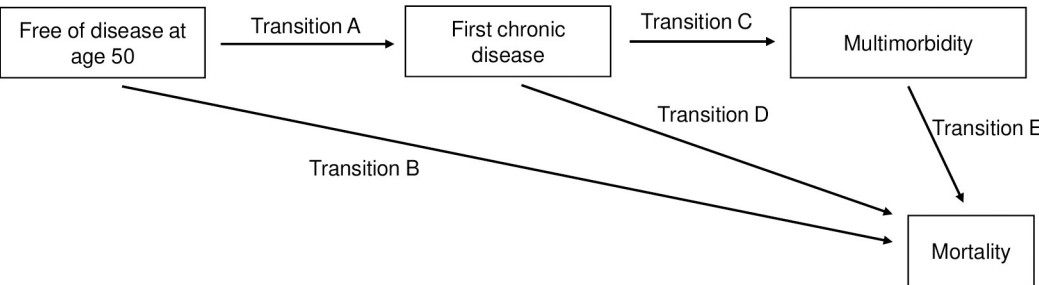

**Fig 1. Schematic representation of the transitions from start of follow-up (free of chronic disease at age 50) to a first chronic disease, multimorbidity, and mortality.** Transition A represents the transition from a healthy state at age 50 (free of the 13 chronic diseases considered) to a first chronic disease (any from the list of 13 diseases considered); Transition B represents the transition from a healthy state to death among those who remained free from any of the 13 diseases during follow-up; Transition C represents the transition from a first chronic disease to multimorbidity (occurrence of a second disease among those with 1 chronic disease); Transition D represents the transition from a first chronic disease to death among those who remained free from multimorbidity during the follow-up; and Transition E represents the transition from multimorbidity to death.

## Results

### Sleep duration at ages 50, 60, and 70 and subsequent risk of multimorbidity

Among the 10,308 participants of the Whitehall cohort, 7,864 (32.5% women) participants free of multimorbidity had data on sleep duration and covariates at age 50 (mean (standard deviation (SD)) = 50.6 (2.6) years). Among them, 2,659 (33.8%) developed multimorbidity at mean age 70.9 (SD = 7.7) years over a mean follow-up of 22.6 (SD = 7.5) years. Among the 6,848 participants with data on sleep duration and covariates at age 60 (mean (SD) = 60.3 (2.2) years) and free of multimorbidity, 2,029 (29.6%) developed multimorbidity at mean age of 72.0 (SD = 6.3) years over a mean follow-up of 13.4 (SD = 6.0) years. Among 5,546 participants free of multimorbidity and with data on sleep duration and covariates at age 70 (mean (SD) = 69.2 (1.9) years), 1,402 (25.3%) subsequently developed multimorbidity at a mean age of 76.0 (SD = 4.8) years over a mean follow-up of 6.8 (SD = 4.5) years. Flowchart of sample selection is shown in Fig 2. Characteristics of participants at age 50 are presented in Table 1 and at age 60 and 70 in S1 and S2 Tables, respectively. At age 50, 544 (6.9% of the study population, N = 7,864) reported sleeping ≤5 hours, 2,562 (32.6%) 6 hours, 3,589 (45.6%) 7 hours, 1,092 (13.9%) 8 hours, and 77 (1.0%) ≥9 hours.

By design, mean age at multimorbidity onset was higher in analyses of sleep duration at older ages, but the distribution of chronic disease dyads was similar across analyses of sleep duration at 50, 60, or 70 years of age (S3 Table). The 8 most common dyads were the same in these analyses, and they represented over 50% of cases of incident multimorbidity cases in all 3 analyses. Coronary heart disease was present in 5 dyads; diabetes, cancer, and arthritis/rheumatoid arthritis in 3; and depression and heart failure in 1 of these 8 dyads.

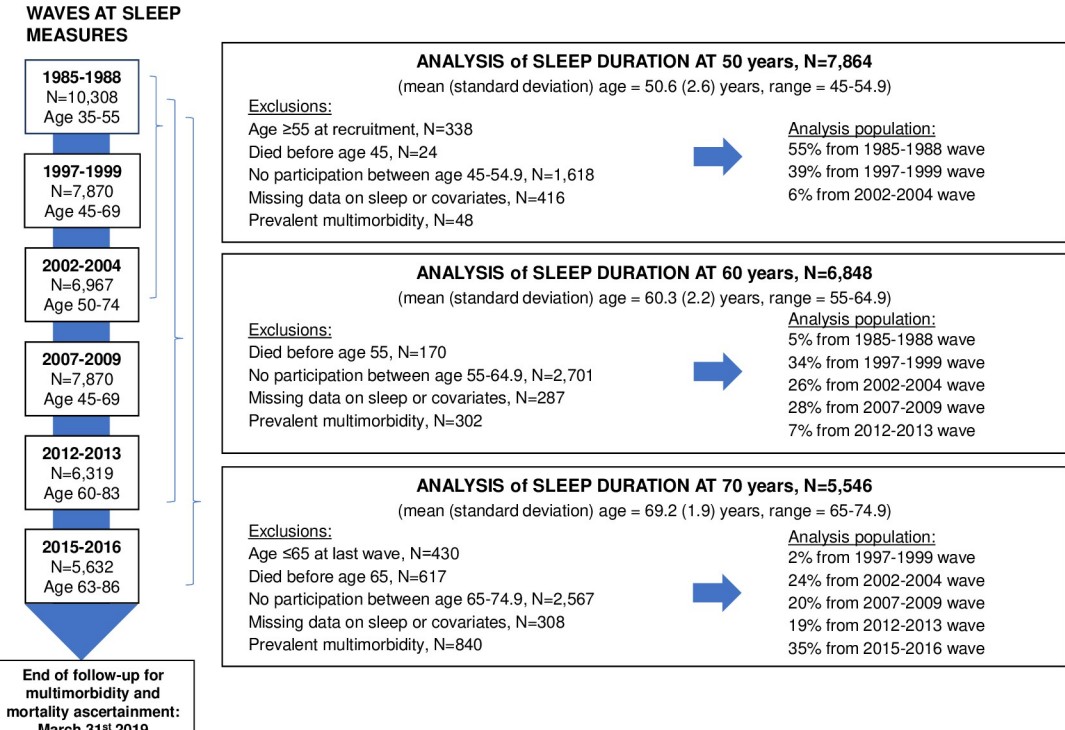

**Fig 2. Flowchart for analyses on the association between sleep duration at age 50, 60, and 70 and risk of multimorbidity.**

**Table 1. Characteristics of the study population at age 50.**

| | Total | ≤5 hours | 6 hours | 7 hours | 8 hours | ≥9 hours | P |
|---|---|---|---|---|---|---|---|
| | | | | Sleep duration at age 50 | | | |
| N | 7,864 | 544 | 2,562 | 3,589 | 1,092 | 77 | |
| Sex | | | | | | | <0.001 |
| Men | 5,305 (67.5) | 319 (58.6) | 1,773 (69.2) | 2,491 (69.4) | 683 (62.5) | 39 (50.6) | |
| Women | 2,559 (32.5) | 225 (41.4) | 789 (30.8) | 1,098 (30.6) | 409 (37.5) | 38 (49.4) | |
| Ethnicity | | | | | | | <0.001 |
| White | 7,088 (90.1) | 464 (85.3) | 2,326 (90.8) | 3,286 (91.6) | 956 (87.5) | 56 (72.7) | |
| Non-white | 776 (9.9) | 80 (14.7) | 236 (9.2) | 303 (8.4) | 136 (12.5) | 21 (27.3) | |
| Education | | | | | | | 0.017 |
| Primary school or less | 887 (11.3) | 81 (14.9) | 288 (11.2) | 391 (10.9) | 121 (11.1) | 6 (7.8) | |
| Lower secondary school | 2,875 (36.6) | 217 (39.9) | 909 (35.5) | 1,312 (36.6) | 412 (37.7) | 25 (32.5) | |
| Higher secondary school | 1,982 (25.2) | 135 (24.8) | 681 (26.6) | 878 (24.5) | 262 (24.0) | 26 (33.8) | |
| University | 1,598 (20.3) | 85 (15.6) | 521 (20.3) | 764 (21.3) | 212 (19.4) | 16 (20.8) | |
| Higher degree | 522 (6.6) | 26 (4.8) | 163 (6.4) | 244 (6.8) | 85 (7.8) | 4 (5.2) | |
| Occupational position | | | | | | | <0.001 |
| Low | 1,572 (20.0) | 153 (28.1) | 480 (18.7) | 658 (18.3) | 258 (23.6) | 23 (29.9) | |
| Intermediate | 3,373 (42.9) | 253 (46.5) | 1,135 (44.3) | 1,517 (42.3) | 432 (39.6) | 36 (46.8) | |
| High | 2,919 (37.1) | 138 (25.4) | 947 (37.0) | 1,414 (39.4) | 402 (36.8) | 18 (23.4) | |
| Marital status | | | | | | | <0.001 |
| Married/cohabiting | 5,951 (75.7) | 344 (63.2) | 1,925 (75.1) | 2,797 (77.9) | 824 (75.5) | 61 (79.2) | |
| Single/divorced/widowed | 1,913 (24.3) | 200 (36.8) | 637 (24.9) | 792 (22.1) | 268 (24.5) | 16 (20.8) | |
| Smoking status | | | | | | | 0.012 |
| Never smoker | 3,885 (49.4) | 247 (45.4) | 1,272 (49.6) | 1,745 (48.6) | 574 (52.6) | 47 (61.0) | |
| Ex-smoker | 2,763 (35.1) | 192 (35.3) | 883 (34.5) | 1,303 (36.3) | 361 (33.1) | 24 (31.2) | |
| Current smoker | 1,216 (15.5) | 105 (19.3) | 407 (15.9) | 541 (15.1) | 157 (14.4) | 6 (7.8) | |
| Alcohol consumption | | | | | | | <0.001 |
| 0 unit/week | 1,399 (17.8) | 128 (23.5) | 419 (16.4) | 590 (16.4) | 238 (21.8) | 24 (31.2) | |
| 1–14 units/week | 4,310 (54.8) | 271 (49.8) | 1,407 (54.9) | 2,024 (56.4) | 572 (52.4) | 36 (46.8) | |
| >14 units/week | 2,155 (27.4) | 145 (26.7) | 736 (28.7) | 975 (27.2) | 282 (25.8) | 17 (22.1) | |
| Fruit and vegetable consumption | | | | | | | 0.010 |
| Less than once a day | 2,857 (36.3) | 233 (42.8) | 950 (37.1) | 1,271 (35.4) | 377 (34.5) | 26 (33.8) | |
| Once a day | 3,025 (38.5) | 199 (36.6) | 940 (36.7) | 1,412 (39.3) | 446 (40.8) | 28 (36.4) | |
| Twice or more a day | 1,982 (25.2) | 112 (20.6) | 672 (26.2) | 906 (25.2) | 269 (24.6) | 23 (29.9) | |
| Moderate-to-vigorous physical activity (hours), M(SD) | 3.3 (3.7) | 2.9 (4.0) | 3.3 (3.9) | 3.4 (3.5) | 3.3 (3.6) | 2.4 (3.1) | 0.009 |
| BMI (kg/m$^2$), M(SD) | 25.5 (3.8) | 26.3 (4.8) | 25.8 (3.9) | 25.3 (3.7) | 25.1 (3.6) | 25.8 (3.5) | <0.001 |
| <18.5 kg/m$^2$ | 78 (1.0) | 8 (1.5) | 18 (0.7) | 35 (1.0) | 16 (1.5) | 1 (1.3) | <0.001 |
| 18.5–24.9 kg/m$^2$ | 3,906 (49.7) | 232 (42.6) | 1,185 (46.3) | 1,882 (52.4) | 575 (52.7) | 32 (41.6) | |
| 25–29.9 kg/m$^2$ | 3,006 (38.2) | 201 (36.9) | 1,054 (41.1) | 1,323 (36.9) | 394 (36.1) | 34 (44.2) | |
| ≥30 kg/m$^2$ | 874 (11.1) | 103 (18.9) | 305 (11.9) | 349 (9.7) | 107 (9.8) | 10 (13.0) | |
| Hypertension | 1,816 (23.1) | 150 (27.6) | 613 (23.9) | 774 (21.6) | 259 (23.7) | 20 (26.0) | 0.014 |
| Use of sleep medication | 107 (1.4) | 20 (3.7) | 47 (1.8) | 33 (0.9) | 5 (0.5) | 2 (2.6) | <0.001 |
| Prevalence of one chronic disease[a] at age 50 | 647 (8.2) | 70 (12.9) | 212 (8.3) | 266 (7.4) | 84 (7.7) | 15 (19.5) | <0.001 |

[a] Chronic disease among diabetes, cancer, coronary heart disease, stroke, heart failure, chronic obstructive pulmonary disease, chronic kidney disease, liver disease, depression, dementia, other mental disorder, Parkinson's disease, and arthritis/rheumatoid arthritis.

Values are No. (%) unless stated otherwise.

BMI, body mass index; M, mean; SD, standard deviation.

**Table 2. Association of sleep duration at age 50, 60, and 70 with risk of multimorbidity[a].**

| | N cases/N total | Model 1: Unadjusted model (age as timescale) | | Model 2: Adjusted for sociodemographic variables[b] | | Model 3: Model 2 + behavioral and health-related factors[c] | |
|---|---|---|---|---|---|---|---|
| | | HR (95% CI) | *p*-value | HR (95% CI) | *p*-value | HR (95% CI) | *p*-value |
| **Sleep duration at age 50** | N cases/N total = 2,659/7,864; follow-up mean (SD) = 22.6 (7.5) years; mean age at event (SD) = 70.9 (7.7) years | | | | | | |
| ≤5 hours | 225/544 | 1.57 (1.36, 1.81) | <0.001 | 1.46 (1.27, 1.69) | <0.001 | 1.30 (1.12, 1.50) | <0.001 |
| 6 hours | 852/2,562 | 1.13 (1.03, 1.23) | 0.007 | 1.11 (1.02, 1.21) | 0.020 | 1.08 (0.98, 1.17) | 0.109 |
| 7 hours | 1,184/3,589 | 1.00 (ref) | | 1.00 (ref) | | 1.00 (ref) | |
| 8 hours | 365/1,092 | 1.01 (0.90, 1.14) | 0.811 | 0.99 (0.88, 1.12) | 0.918 | 1.00 (0.89, 1.12) | 0.953 |
| ≥9 hours | 33/77 | 1.60 (1.13, 2.27) | 0.008 | 1.42 (1.00, 2.02) | 0.047 | 1.39 (0.98, 1.96) | 0.067 |
| **Sleep duration at age 60** | N cases/N total = 2,029/6,848; follow-up mean (SD) = 13.4 (6.0) years; mean age at event (SD) = 72.0 (6.3) years | | | | | | |
| ≤5 hours | 202/519 | 1.56 (1.34, 1.83) | <0.001 | 1.44 (1.23, 1.68) | <0.001 | 1.32 (1.13, 1.55) | 0.001 |
| 6 hours | 645/2,095 | 1.16 (1.04, 1.28) | 0.006 | 1.13 (1.02, 1.26) | 0.018 | 1.14 (1.02, 1.26) | 0.016 |
| 7 hours | 793/2,882 | 1.00 (ref) | | 1.00 (ref) | | 1.00 (ref) | |
| 8 hours | 340/1,230 | 1.03 (0.91, 1.17) | 0.652 | 1.02 (0.90, 1.16) | 0.715 | 1.06 (0.93, 1.20) | 0.375 |
| ≥9 hours | 49/122 | 1.64 (1.23, 2.19) | 0.001 | 1.60 (1.20, 2.14) | 0.001 | 1.54 (1.15, 2.06) | 0.003 |
| **Sleep duration at age 70** | N cases/N total = 1,402/5,546; follow-up mean (SD) = 6.8 (4.5) years; mean age at event (SD) = 76.0 (4.8) years | | | | | | |
| ≤5 hours | 152/451 | 1.63 (1.36, 1.95) | <0.001 | 1.58 (1.31, 1.90) | <0.001 | 1.40 (1.16, 1.68) | <0.001 |
| 6 hours | 425/1,574 | 1.20 (1.05, 1.36) | 0.007 | 1.18 (1.04, 1.34) | 0.013 | 1.12 (0.98, 1.27) | 0.092 |
| 7 hours | 514/2,249 | 1.00 (ref) | | 1.00 (ref) | | 1.00 (ref) | |
| 8 hours | 269/1,151 | 1.04 (0.90, 1.21) | 0.595 | 1.03 (0.89, 1.20) | 0.649 | 0.99 (0.85, 1.14) | 0.843 |
| ≥9 hours | 42/121 | 1.52 (1.11, 2.08) | 0.010 | 1.48 (1.08, 2.03) | 0.015 | 1.51 (1.10, 2.08) | 0.010 |

[a] Multimorbidity defined as 2 or more of the following chronic diseases: diabetes, cancer, coronary heart disease, stroke, heart failure, chronic obstructive pulmonary disease, chronic kidney disease, liver disease, depression, dementia, other mental disorder, Parkinson's disease, and arthritis/rheumatoid arthritis.
[b] Adjusted for age (timescale), sex, ethnicity, education, occupational position, and marital status.
[c] Additionally adjusted for alcohol consumption, physical activity, smoking status, fruit and vegetable consumption, BMI, hypertension, use of sleep medication, and prevalence of 1 of the 13 chronic diseases.
CI, confidence intervals; HR, hazard ratio; ref, reference; SD, standard deviation.

Table 2 shows the association of sleep duration at age 50, 60, and 70 with subsequent risk of multimorbidity. In the absence of sex differences (p for interaction between sex and sleep duration >0.05), men and women were combined in the analyses. In analyses adjusted for sociodemographic variables, the risk of multimorbidity was higher in participants with a sleep duration ≤5 hours, 6 hours, and ≥9 hours compared to sleep duration of 7 hours, irrespective of the age at measurement of sleep duration. Further adjustment for health behaviors, BMI, hypertension, use of sleep medication, and prevalence of 1 of the 13 chronic diseases showed

sleep duration ≤5 hours at age 50 (hazard ratio (HR) = 1.30, 95% confidence interval, 1.12 to 1.50; $p < 0.001$), age 60 (HR = 1.32, 1.13 to 1.55; $p < 0.001$), and at age 70 (HR = 1.40, 1.16 to 1.68; $p < 0.001$) to be associated with higher risk of multimorbidity. In these analyses, the association of sleep duration ≥9 hours at age 50 with incident multimorbidity did not reach statistical significance (HR = 1.39, 0.98 to 1.96; $p = 0.067$), while sleep duration ≥9 hours at age 60 (HR = 1.54, 1.15 to 2.06; $p = 0.003$) and 70 (HR = 1.51, 1.10 to 2.08; $p = 0.010$) was associated with higher risk of multimorbidity.

In sensitivity analyses excluding participants with any of the 13 chronic diseases at the measure of sleep duration (S4 Table), the pattern of associations was similar to the main analyses for sleep duration at age 50, 60, and 70 years apart from the association with sleep duration ≥9 hours at age 70 that did not reach significance although the effect size was comparable (HR = 1.52, 0.98 to 2.35; $p = 0.063$). Analyses excluding participants on sleep medication were similar, irrespective of age at sleep measure (S5 Table).

Accelerometer data and covariates were available on 3,920 participants who took part to the accelerometer sub-study at the 2012 wave. Of them, 3,368 participants (mean age (SD) = 68.9 (5.6), range = 60 to 83 years) were free of multimorbidity and were included in the analysis. During a mean follow-up of 6 years, 601 developed multimorbidity. The correlation between accelerometer- and questionnaire-assessed sleep duration was moderate (Pearson correlation = 0.41, $p < 0.001$). The shape of the association between accelerometer-assessed sleep duration and incident multimorbidity was similar to that observed with self-reported sleep duration, with the lowest risk of incident multimorbidity seen at 7 hours of sleep (Fig 3). Given the small number of incident multimorbidity cases in participants in less than 6 hours and more than 8 hours categories (Fig 3, panel D), the focus here was on the shape of the association rather than estimates of the associations.

In post hoc analysis, first we showed results from analyses using inverse probability weighting to account for missing data to be consistent with those in the main analysis (S6 Table). Second, analysis excluding 1 disease at a time from the definition of multimorbidity showed results to be robust to the list of chronic diseases used to define multimorbidity (S7 Table). Third, we examined trajectories of sleep duration. The Pearson correlation of sleep duration at age 50 with sleep duration at age 60 was 0.49 and with sleep duration at age 70 was 0.42 ($p < 0.001$). A moderate correlation was also found between sleep duration at age 60 and age 70 (correlation 0.57, $p < 0.001$). Six trajectories of sleep duration using data on 5,510 participants were used and labeled as persistent short sleep, persistent normal sleep, persistent long sleep, change from short to normal sleep, change from normal to long sleep, and change from normal to short sleep (S8 Table). Compared to persistent normal sleep, persistent short sleep duration between age 50 and 70 was associated with increased risk of incident multimorbidity (HR = 1.17, 1.01 to 1.35; $p = 0.040$) in analyses adjusted for sociodemographic, behavioral, and health-related factors (S9 Table).

In a fourth post hoc analysis, sleep disturbances at age 60 and 70, measured using the Jenkins sleep problems scale, were examined and found to be greater in those with sleep duration ≤5 hours at age 60 as compared to with 7 hours of sleep (mean (SD) Jenkins sleep problems score = 10.1 (SD = 5.7) versus 4.4 (SD = 3.5); $p < 0.001$; S10 Table) but not in those with sleep duration ≥9 hours (4.4 (SD = 4.1) versus 4.4 (3.5); $p = 0.925$). A similar pattern was observed at age 70 (S10 Table). Greater sleep disturbance at age 60 and at age 70 (HR = 1.03, 1.02 to 1.04 per 1-point increase in the score at both ages; $p < 0.001$) was associated with increased risk of multimorbidity in analyses adjusted for sociodemographic, behavioral, and health-related factors (S11 Table).

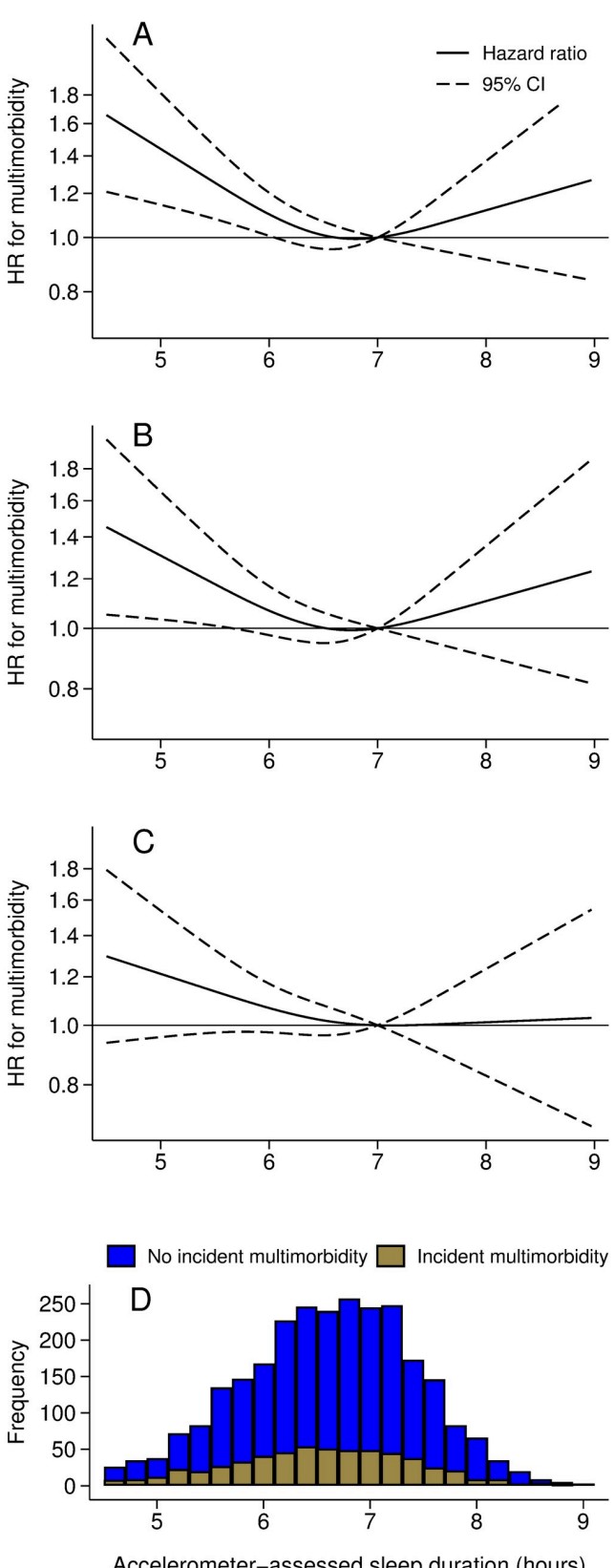

**Fig 3. Association of accelerometer assessed sleep duration in 2012–2013 (age range, 60 to 83 years) with risk of incident multimorbidity (N cases/N total = 601/3,368) over a mean follow-up of 6.0 (SD = 1.6) years.** Multimorbidity defined as 2 or more of the following chronic diseases: diabetes, cancer, coronary heart disease, stroke, heart failure, chronic obstructive pulmonary disease, chronic kidney disease, liver disease, depression, dementia, other mental disorder, Parkinson's disease, and arthritis/rheumatoid arthritis. (A) Model unadjusted (age as timescale). (B) Model adjusted for age (timescale), sex, ethnicity, education, occupational position, and marital status. (C) Model additionally adjusted for alcohol consumption, physical activity, smoking status, fruit and vegetable consumption, BMI, hypertension, use of sleep medication, and prevalence of 1 of the 13 chronic diseases. (D) Sleep duration distribution among participants with no incident multimorbidity (blue) and those with incident multimorbidity (brown).

## Sleep duration at age 50 and subsequent transitions to a first chronic disease, multimorbidity, and death

A total of 7,217 participants with data on sleep duration at age 50 were free from the 13 chronic diseases considered in this study. Among them, over a mean follow-up of 25.2 years, 213 participants died without having developed any of the 13 diseases, 4,446 participants developed 1 chronic disease and of them 2,297 subsequently developed a second disease (multimorbidity), and of them 787 died. In analysis adjusted for sociodemographic, behavioral, and health-related factors, compared to sleep duration of 7 hours, sleep duration ≤5 hours was associated with increased risk of transition to a first chronic disease (HR = 1.20, 1.06 to 1.35; $p$ = 0.003) and subsequent transition to multimorbidity (HR = 1.21, 1.03 to 1.42; $p$ = 0.021) but not mortality (Table 3). Sleep durations longer than 7 hours were not associated with these transitions.

Similar findings were observed in inverse probability weighting analyses to account for missing data (S12 Table). In post hoc analysis, we also examined the association between sleep duration at age 50 and risk of mortality without considering chronic diseases over the follow-up; in analysis adjusted for sociodemographic, behavioral, and health-related factors, sleep duration ≤5 hours (HR = 1.25, 1.02 to 1.53; $p$ = 0.034) and of 6 hours (HR = 1.17, 1.04 to 1.32; $p$ = 0.008) was associated with higher risk of mortality over a mean follow-up of 25.2 (SD = 6.9) years (S13 Table).

## Sleep duration after a first chronic disease and subsequent risk of multimorbidity and death

A total of 6,546 participants were diagnosed with 1 of the 13 chronic diseases considered during the follow-up period, constituting the target population of this analysis. Among them, 2,442 did not have data on sleep duration after the onset of this first chronic disease, 464 had data but only after the onset of multimorbidity, and 42 had missing covariates, leading to an analytical sample of 3,702 participants (S1 Fig). In analysis adjusted for sociodemographic, behavioral, and health-related factors, sleep duration ≤5 hours in this sample was associated with higher risk of incident multimorbidity (HR = 1.20, 1.03 to 1.40; $p$ = 0.018) (Table 4). Sleep duration ≥9 hours was associated with a higher risk of multimorbidity (HR = 1.46, 1.07 to 1.99; $p$ = 0.017) in analysis adjusted for sociodemographic factors, although the association was attenuated after adjustment for behavioral and health-related factors (HR = 1.36, 1.00 to 1.86; $p$ = 0.051). No consistent association was found with transition to mortality. Once missing data were taken into account using inverse-probability weighting (S14 Table), findings remain substantially the same.

## Discussion

This prospective study spanning over 20 years presents 3 key findings. One, short sleep duration was consistently associated with increased risk of multimorbidity, irrespective of sleep

**Table 3. Association of sleep duration at age 50 with transitions from a healthy state to first chronic disease, multimorbidity, and mortality (*N* = 7,217).**

| Sleep duration at 50y | N cases/N total | Model 1: Unadjusted model (age as timescale) | | Model 2: Adjusted for sociodemographic variables[a] | | Model 3: Model 2 + behavioral and health-related factors[b] | |
|---|---|---|---|---|---|---|---|
| | | HR (95% CI) | *p*-value | HR (95% CI) | *p*-value | HR (95% CI) | *p*-value |
| **Healthy to first chronic disease[c], transition A (N cases/N total = 4,446/7,217; mean age at event (SD) = 66.2 (8.5) years)** | | | | | | | |
| ≤5 hours | 319/474 | 1.31 (1.16, 1.47) | <0.001 | 1.28 (1.14, 1.44) | <0.001 | 1.20 (1.06, 1.35) | 0.003 |
| 6 hours | 1,428/2,350 | 1.07 (1.00, 1.14) | 0.059 | 1.06 (0.99, 1.13) | 0.097 | 1.03 (0.96, 1.10) | 0.418 |
| 7 hours | 2,038/3,323 | 1.00 (ref) | | 1.00 (ref) | | 1.00 (ref) | |
| 8 hours | 621/1,008 | 1.00 (0.91, 1.09) | 0.944 | 0.99 (0.90, 1.08) | 0.745 | 0.98 (0.90, 1.08) | 0.730 |
| ≥9 hours | 40/62 | 1.16 (0.84, 1.58) | 0.367 | 1.10 (0.80, 1.50) | 0.558 | 1.10 (0.80, 1.50) | 0.571 |
| **Healthy to death, transition B (N cases/N total = 213/7,217; mean age at event (SD) = 65.2 (9.0) years)** | | | | | | | |
| ≤5 hours | 14/474 | 1.24 (0.71, 2.18) | 0.447 | 1.12 (0.64, 1.97) | 0.689 | 0.97 (0.55, 1.72) | 0.926 |
| 6 hours | 71/2,350 | 1.17 (0.86, 1.59) | 0.314 | 1.12 (0.82, 1.52) | 0.465 | 1.06 (0.78, 1.44) | 0.727 |
| 7 hours | 98/3,323 | 1.00 (ref) | | 1.00 (ref) | | 1.00 (ref) | |
| 8 hours | 29/1,008 | 0.97 (0.64, 1.47) | 0.902 | 0.96 (0.63, 1.45) | 0.829 | 0.96 (0.63, 1.45) | 0.844 |
| ≥9 hours | 1/62 | na | | na | | na | |
| **First chronic disease[c] to multimorbidity[c], transition C (N cases/N total = 2,297/4,446; mean age at event (SD) = 71.9 (7.2) years)** | | | | | | | |
| ≤5 hours | 183/319 | 1.34 (1.15, 1.57) | 0.000 | 1.27 (1.09, 1.50) | 0.003 | 1.21 (1.03, 1.42) | 0.021 |
| 6 hours | 739/1,428 | 1.13 (1.02, 1.24) | 0.014 | 1.12 (1.02, 1.23) | 0.019 | 1.10 (1.00, 1.21) | 0.041 |
| 7 hours | 1,042/2,038 | 1.00 (ref) | | 1.00 (ref) | | 1.00 (ref) | |
| 8 hours | 310/621 | 0.98 (0.86, 1.11) | 0.768 | 0.98 (0.86, 1.11) | 0.767 | 0.98 (0.86, 1.11) | 0.771 |
| ≥9 hours | 23/40 | 1.29 (0.85, 1.95) | 0.226 | 1.18 (0.77, 1.79) | 0.451 | 1.10 (0.72, 1.68) | 0.661 |
| **First chronic disease[c] to death, transition D (N cases/N total = 474/4,446; mean age at event (SD) = 68.9 (7.9) years)** | | | | | | | |
| ≤5 hours | 32/319 | 1.17 (0.80, 1.69) | 0.422 | 1.16 (0.79, 1.69) | 0.454 | 1.19 (0.81, 1.74) | 0.374 |
| 6 hours | 163/1,428 | 1.27 (1.03, 1.56) | 0.023 | 1.27 (1.04, 1.57) | 0.022 | 1.30 (1.06, 1.60) | 0.013 |
| 7 hours | 205/2,038 | 1.00 (ref) | | 1.00 (ref) | | 1.00 (ref) | |
| 8 hours | 69/621 | 1.09 (0.83, 1.43) | 0.543 | 1.09 (0.83, 1.44) | 0.516 | 1.10 (0.83, 1.44) | 0.509 |
| ≥9 hours | 5/40 | na | | na | | na | |
| **Multimorbidity[c] to death, transition E (N cases/N total = 787/2,297; mean age at event (SD) = 76.0 (6.8) years)** | | | | | | | |
| ≤5 hours | 65/183 | 1.04 (0.79, 1.35) | 0.788 | 1.07 (0.82, 1.40) | 0.636 | 1.07 (0.81, 1.40) | 0.631 |
| 6 hours | 254/739 | 1.09 (0.93, 1.28) | 0.290 | 1.10 (0.94, 1.30) | 0.236 | 1.10 (0.94, 1.30) | 0.236 |
| 7 hours | 348/1,042 | 1.00 (ref) | | 1.00 (ref) | | 1.00 (ref) | |

(*Continued*)

**Table 3.** (Continued)

| Sleep duration at 50y | N cases/N total | Model 1: Unadjusted model (age as timescale) | | Model 2: Adjusted for sociodemographic variables[a] | | Model 3: Model 2 + behavioral and health-related factors[b] | |
|---|---|---|---|---|---|---|---|
| 8 hours | 112/310 | 1.06 (0.85, 1.31) | 0.613 | 1.08 (0.87, 1.34) | 0.462 | 1.09 (0.88, 1.35) | 0.420 |
| ≥9 hours | 8/23 | 0.93 (0.46, 1.88) | 0.843 | 1.10 (0.54, 2.24) | 0.785 | 1.09 (0.54, 2.22) | 0.808 |

[a] Adjusted for age (timescale), sex, ethnicity, education, occupational position, and marital status.

[b] Additionally adjusted for alcohol consumption, physical activity, smoking status, fruit and vegetable consumption, BMI, hypertension, and use of sleep medication.

[c] Chronic disease among diabetes, cancer, coronary heart disease, stroke, heart failure, chronic obstructive pulmonary disease, chronic kidney disease, liver disease, depression, dementia, other mental disorder, Parkinson's disease, and arthritis/rheumatoid arthritis; multimorbidity defined as 2 or more of these diseases.

For transitions see Fig 1.

CI, confidence intervals; HR, hazard ratio; na, not applicable (N cases ≤5); ref, reference; SD, standard deviation.

being measured in mid or late life. The analysis of transitions in health states showed short sleep to be associated with the onset of a first disease and subsequent multimorbidity but not disease prognosis, measured using mortality. Two, the results for long sleep duration were less robust as associations with multimorbidity were observed when sleep was measured at age 60 and 70 but not at 50 years. In the analyses of transitions in health states, we also found long sleep at age 50 not to be associated with disease progression although some of the transitions could not be examined due to a small number of cases. Three, the accelerometer-based measure of sleep duration, undertaken when the mean age of participants was 69 years (range 60 to 83), confirmed the shape of the association between sleep duration and multimorbidity in the main analysis with results matching those observed for self-reported sleep duration at ages 60 or 70. Taken together, these findings suggest an association between short sleep duration and development of multimorbidity.

Much of the evidence on the role of sleep duration for health comes from studies on individual chronic diseases, rather than multimorbidity. One meta-analysis suggested that the association between short sleep duration and health outcomes is stronger when sleep duration is assessed before age 65 [16] and another found long sleep duration, particularly at older ages, to be more strongly associated with chronic diseases [19]. While these studies are important, their generalizability to real-life settings is limited as most adults live with multiple rather than single chronic conditions [8]. Apart from 1 notable study that examined the association between sleep disturbance (rather than sleep duration) and number of chronic conditions over a 9-year period [31], the few studies that exist on sleep duration and multimorbidity are cross-sectional [9–13]. These studies report a U-shape association between sleep duration and multimorbidity, implying higher prevalence of multimorbidity in both individuals with short and long sleep duration [9–13]. In some studies, stronger association was found for short sleep [10,12,13] while at least 1 study suggested a stronger link with long sleep [11]. Although informative, these cross-sectional findings might reflect both the impact of sleep duration on disease incidence and the converse, that is, the effect of disease on sleep.

Using a prospective design, our results show robust evidence of an association of short sleep duration with incident multimorbidity; this was the case for sleep duration measured either in mid or late life or trajectories of sleep duration between age 50 and 70. These findings were supported by results of the multistate models where short sleep duration at age 50 was

**Table 4. Association of sleep duration after a first chronic disease with transitions from first chronic disease to multimorbidity and mortality (N = 3,702).**

| Sleep duration after a first chronic disease | N cases/N total | Model 1: Unadjusted model (age as timescale) | | Model 2: Adjusted for sociodemographic variables[a] | | Model 3: Model 2 + behavioral and health-related factors[b] | |
|---|---|---|---|---|---|---|---|
| | | HR (95% CI) | p-value | HR (95% CI) | p-value | HR (95% CI) | p-value |
| **First chronic disease[c] to multimorbidity[c] (N cases/N total = 1,923/3,702; mean age at event (SD) = 70.1 (7.9) years)** | | | | | | | |
| ≤5 hours | 226/385 | 1.33 (1.15, 1.55) | <0.001 | 1.25 (1.08, 1.46) | 0.004 | 1.20 (1.03, 1.40) | 0.018 |
| 6 hours | 588/1,125 | 1.11 (0.99, 1.24) | 0.066 | 1.10 (0.98, 1.22) | 0.105 | 1.08 (0.97, 1.20) | 0.179 |
| 7 hours | 713/1,440 | 1.00 (ref) | | 1.00 (ref) | | 1.00 (ref) | |
| 8 hours | 353/685 | 1.09 (0.96, 1.23) | 0.206 | 1.08 (0.95, 1.23) | 0.244 | 1.08 (0.95, 1.23) | 0.242 |
| ≥9 hours | 43/67 | 1.50 (1.10, 2.04) | 0.010 | 1.46 (1.07, 1.99) | 0.017 | 1.36 (1.00, 1.86) | 0.051 |
| **First chronic disease[c] to death (N cases/N total = 190/3,702; mean age at event (SD) = 69.7 (8.5) years)** | | | | | | | |
| ≤5 hours | 20/385 | 1.28 (0.77, 2.10) | 0.338 | 1.23 (0.74, 2.05) | 0.420 | 1.23 (0.74, 2.05) | 0.432 |
| 6 hours | 58/1,125 | 1.16 (0.81, 1.64) | 0.414 | 1.13 (0.79, 1.61) | 0.497 | 1.11 (0.78, 1.58) | 0.564 |
| 7 hours | 68/1,440 | 1.00 (ref) | | 1.00 (ref) | | 1.00 (ref) | |
| 8 hours | 40/685 | 1.30 (0.88, 1.92) | 0.192 | 1.32 (0.89, 1.96) | 0.164 | 1.34 (0.90, 1.99) | 0.146 |
| ≥9 hours | 4/67 | na | | na | | na | |
| **Multimorbidity[c] to death (N cases/N total = 545/1,923; mean age at event (SD) = 74.9 (7.5) years)** | | | | | | | |
| ≤5 hours | 63/226 | 1.08 (0.81, 1.43) | 0.608 | 1.14 (0.85, 1.52) | 0.382 | 1.17 (0.87, 1.57) | 0.292 |
| 6 hours | 158/588 | 0.98 (0.80, 1.21) | 0.867 | 1.01 (0.82, 1.25) | 0.929 | 0.99 (0.80, 1.23) | 0.922 |
| 7 hours | 194/713 | 1.00 (ref) | | 1.00 (ref) | | 1.00 (ref) | |
| 8 hours | 121/353 | 1.33 (1.06, 1.67) | 0.015 | 1.39 (1.10, 1.75) | 0.005 | 1.41 (1.12, 1.78) | 0.003 |
| ≥9 hours | 9/43 | 0.70 (0.36, 1.37) | 0.295 | 0.67 (0.34, 1.31) | 0.243 | 0.70 (0.36, 1.37) | 0.294 |

[a] Adjusted for age (timescale), sex, ethnicity, education, occupational position, and marital status.

[b] Additionally adjusted for alcohol consumption, physical activity, smoking status, fruit and vegetable consumption, BMI, hypertension, and use of sleep medication.

[c] Chronic disease among diabetes, cancer, coronary heart disease, stroke, heart failure, chronic obstructive pulmonary disease, chronic kidney disease, liver disease, depression, dementia, other mental disorder, Parkinson's disease, and arthritis/rheumatoid arthritis; multimorbidity defined as 2 or more of these diseases.

CI, confidence intervals; HR, hazard ratio; na, not applicable (N cases ≤5); ref, reference; SD, standard deviation.

associated with higher risk of onset of a first chronic disease and subsequent multimorbidity. Short sleep duration was also associated with greater sleep disturbances—itself associated with increased risk of multimorbidity in the present study as well as in a previous study [31]—suggesting that short sleep duration might be a marker of poor sleep quality. Sleep duration and quality might impact health via their role in regulation of endocrine and metabolic processes, inflammation, and circadian rhythm [15,16]. There was no evidence in the present data that short sleep duration was associated with progression to death among those with existing

chronic disease(s). This suggests that the previously reported association between short sleep duration and mortality [2–4] is likely to be driven by the association of short sleep with onset of chronic diseases that are themselves associated with risk of mortality.

There is some evidence of poorer health outcomes in long sleepers in previous studies [2,19], but the mechanisms underlying this association remain unclear [19]. Long sleep duration has been hypothesized to reflect poor overall sleep quality that could have a detrimental impact on health [15,19,31], although this hypothesis was not supported in our study where the sleep disturbances were similar in those sleeping ≥9 hours and those sleeping 7 hours. It is also possible that long sleep duration is a marker of underlying conditions that are themselves associated with an increased risk of chronic disease and mortality. This hypothesis is supported by studies showing the association of long sleep duration with health to be based primarily on older adults, who are more likely to have preexisting medical conditions [19]. Our analyses provide further support for this hypothesis as the association of long sleep duration with multimorbidity was attenuated when sleep duration was measured in disease-free participants at age 50. We also found long sleep duration after a first disease to be associated with subsequent risk of multimorbidity although this association was partly attributable to behavioral and health-related factors. These findings support the notion that previously reported associations between long sleep duration and health might reflect increased sleep duration among those with existing health conditions, rather than long sleep duration being an important risk factor for disease onset.

A major strength of this study is the long follow-up, repeated measures that allowed analyses on sleep duration at ages 50, 60, and 70 along with sleep duration trajectories over this age range. Compared to conventional analyses that examined associations between sleep duration and health outcomes, the use of multistate models provides additional insight into the association of sleep duration with the course of disease, including the finding that sleep duration is associated with onset of a chronic disease and subsequent multimorbidity but not with mortality among persons with these conditions.

Our findings should also be considered in light of the limitations of the study. First, like most large-scale studies on sleep, we used self-reported sleep which is likely to be subject to reporting bias. Although the correlation between accelerometer-assessed and self-reported sleep duration was moderate in our study, the shape of the association with multimorbidity risk was similar with both measures. Second, data on sleep quality were available only at age 60 and 70. Third, participants from the Whitehall II cohort study were all employed at recruitment to the study and likely to be healthier than the general population. However, the association between risk factors and health outcomes has been shown to be similar to that observed in the general population [40]. Fourth, participants were mostly of white ethnicity, reflecting the population of the country in 1991 [41], and whether results are generalizable to other populations is unknown. Fifth, despite the use of several covariates residual confounding can be an issue in observational studies. For example, short sleep duration and sleep disturbances may reflect the symptoms of undiagnosed diseases at sleep measures such as depression or arthritis. Considering the wide range of covariates, a confounder would need to have a risk ratio of 2 (E-value) with both the exposure and the outcome to explain away the association between sleep duration at age 50 and multimorbidity. Finally, the mortality numbers in the ≥9 hours sleep duration group was small in some of the analyses, not allowing firm conclusions to be drawn on the association between long sleep and mortality.

With population ageing and increases in life expectancy, living with multiple chronic conditions is common among older adults in high-income countries [6,42]. Multimorbidity presents a challenge as it is associated with high health care service use, hospitalizations, and disability; a further concern is that contemporary health care systems are organized around

treatment and care of individual diseases rather than multimorbidity [43]. Primary prevention of a first chronic disease and secondary prevention to reduce risk of multimorbidity among those with a first chronic disease are thus important in addressing the burden of multimorbidity [42]. The present findings along with evidence from previous studies show the importance of sleep duration across the lifecourse for health outcomes at older ages [1–4]. Further research using objective measures of sleep duration would allow better understanding of the importance of sleep duration for chronic disease and multimorbidity.

In conclusions, findings from the present study suggest short sleep duration in midlife and old age is associated with higher risk of onset of chronic disease and multimorbidity. These findings support the promotion of good sleep hygiene in both primary and secondary prevention by targeting behavioral and environmental conditions that affect sleep duration and quality [44].

## Supporting information

**S1 STROBE Checklist. STROBE Checklist.**
(DOCX)

**S1 Text. Analysis plan drafted in August 2021 before data analysis.**
(DOCX)

**S1 Fig. Flowchart for the analyses on the association of sleep duration after a first chronic disease with transitions from first chronic disease to multimorbidity and mortality.**
(TIF)

**S1 Table. Characteristics of the study population at age 60.**
(DOCX)

**S2 Table. Characteristics of the study population at age 70.**
(DOCX)

**S3 Table. Description of chronic disease dyads in the analysis of sleep duration at age 50, 60, and 70 and incident multimorbidity.**
(DOCX)

**S4 Table. Association of sleep duration at age 50, 60, and 70 with risk of multimorbidity among participants free from prevalent chronic disease.**
(DOCX)

**S5 Table. Association of sleep duration at age 50, 60, and 70 with risk of multimorbidity in analyses excluding participants on sleep medication.**
(DOCX)

**S6 Table. Association of sleep duration at age 50, 60, and 70 with risk of multimorbidity using inverse probability weighting analyses to take missing data into account.**
(DOCX)

**S7 Table. Association of sleep duration at age 50, 60, and 70 with risk of multimorbidity: Impact of removing 1 chronic disease at a time from the list of chronic diseases included in the definition of multimorbidity.**
(DOCX)

**S8 Table. Description of sleep duration at age 50, 60, and 70 by groups of trajectories of sleep duration between age 50 and 70.**
(DOCX)

**S9 Table. Association of trajectories of sleep duration between age 50 and 70 with risk of multimorbidity.**
(DOCX)

**S10 Table. Sleep disturbances assessed using the Jenkins sleep problems scale as a function of sleep duration.**
(DOCX)

**S11 Table. Association of the Jenkins sleep problems scale with risk of multimorbidity.**
(DOCX)

**S12 Table. Association of sleep duration at age 50 with transitions from a healthy state to first chronic disease, multimorbidity, and mortality ($N = 7,217$) using inverse probability weighting analyses to take missing data into account.**
(DOCX)

**S13 Table. Association of sleep duration at age 50 with risk of mortality.**
(DOCX)

**S14 Table. Association of sleep duration after first chronic disease with transitions to multimorbidity and mortality in analyses using inverse probability weighting analyses to take missing data into account.**
(DOCX)

## Acknowledgments

We thank all of the participating civil service departments and their welfare, personnel, and establishment officers; the British Occupational Health and Safety Agency; the British Council of Civil Service Unions; all participating civil servants in the Whitehall II study; and all members of the Whitehall II study team. The Whitehall II study team comprises research scientists, statisticians, study coordinators, nurses, data managers, administrative assistants, and data entry staff, who make the study possible.

## Author Contributions

**Conceptualization:** Séverine Sabia, Aline Dugravot, Archana Singh-Manoux.

**Data curation:** Aline Dugravot.

**Formal analysis:** Aline Dugravot.

**Funding acquisition:** Mika Kivimaki, Archana Singh-Manoux.

**Investigation:** Mika Kivimaki, Archana Singh-Manoux.

**Methodology:** Séverine Sabia.

**Supervision:** Archana Singh-Manoux.

**Validation:** Séverine Sabia, Damien Léger, Céline Ben Hassen.

**Visualization:** Aline Dugravot, Céline Ben Hassen.

**Writing – original draft:** Séverine Sabia.

**Writing – review & editing:** Séverine Sabia, Aline Dugravot, Damien Léger, Céline Ben Hassen, Mika Kivimaki, Archana Singh-Manoux.

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
