## [Editor Report · Decision Letter 0]

28 Mar 2022

Dear Dr Sabia, 

Thank you for submitting your manuscript entitled "Sleep duration at age 50, 60, and 70 and risk of multimorbidity: a 25-year follow-up study" for consideration by PLOS Medicine.

Your manuscript has now been evaluated by the PLOS Medicine editorial staff and I am writing to let you know that we would like to send your submission out for external peer review.

Please re-submit your manuscript within two working days, i.e. by Mar 30 2022 11:59PM.

Kind regards,

Caitlin Moyer, Ph.D.

Associate Editor

PLOS Medicine

---

## [Decision Letter · Decision Letter 1]

9 Jun 2022

Dear Dr. Sabia,

Thank you very much for submitting your manuscript "Sleep duration at age 50, 60, and 70 and risk of multimorbidity: a 25-year follow-up study" (PMEDICINE-D-22-00972R1) for consideration at PLOS Medicine. 

Your paper was evaluated by a senior editor and discussed among all the editors here. It was also discussed with an academic editor with relevant expertise, and sent to four independent reviewers, including a statistical reviewer. The reviews are appended at the bottom of this email and any accompanying reviewer attachments can be seen via the link below:

[LINK]

In light of these reviews, I am afraid that we will not be able to accept the manuscript for publication in the journal in its current form, but we would like to consider a revised version that addresses the reviewers' and editors' comments. Obviously we cannot make any decision about publication until we have seen the revised manuscript and your response, and we plan to seek re-review by one or more of the reviewers. 

We expect to receive your revised manuscript by Jun 30 2022 11:59PM. Please email us (plosmedicine@plos.org) if you have any questions or concerns.

We look forward to receiving your revised manuscript. 

Sincerely,

Caitlin Moyer, Ph.D.

Associate Editor

PLOS Medicine

plosmedicine.org

1. Title: Please revise your title according to PLOS Medicine's style. Your title must be nondeclarative and not a question. It should begin with main concept if possible. "Effect of" should be used only if causality can be inferred, i.e., for an RCT. Please place the study design ("A randomized controlled trial," "A retrospective study," "A modelling study," etc.) in the subtitle (ie, after a colon).

2. Data availability statement: The Data Availability Statement (DAS) requires revision. For each data source used in your study:

3. Line numbers: Please provide line numbers, running continuously throughout the document, with the revised version.

4. Abstract: Methods and Findings: Please provide some summary information on the Whitehall II population. Please mention the study dates, and the setting of your study. Please mention the study design.

5. Abstract: Methods and Findings: Please quantify the main results with both 95% CIs and p values. Please ensure that all numbers presented in the abstract are present and identical to numbers presented in the main manuscript text.

6. Abstract: Methods and Findings: In the last sentence of the Abstract Methods and Findings section, please describe the main limitation(s) of the study's methodology.

7. Abstract: Conclusions: Please address the study implications without overreaching what can be concluded from the data; the phrase "In this study, we observed ..." may be useful. Please revise the Conclusions section such that it provides an interpretation of the study based on the results presented in the abstract, emphasizing what is new without overstating your conclusions. For example, the study does not seem to report evidence suggesting that long sleep is a marker of underlying health conditions.

8. Methods: How was ethnicity defined and by whom?

9. Methods: Repeated measures design: Please describe how clustering of repeat measurements at the individual-level was accounted for.

10. Methods: “We repeated this analysis using inverse probability weighting to account for missing data…” Please provide more information/a table describing the nature and amount of missing data.

11. Methods: “...(3) by the association between accelerometer-assessed sleep duration at mean age 69 years and incident multimorbidity. The accelerometer measure and covariates were drawn from the 2012-2013 wave of data collection. Given the detailed data on sleep duration extracted from the accelerometer, we used restricted cubic spline regressions with Harrell knots [32], Stata command xblc [33], with 7-hour sleep as the reference to examine the shape of the association between sleep duration and dementia risk.” Please clarify if the accelerometer data were available for participants at other ages, as well. Please clarify the reason why the association between sleep duration and dementia is highlighted in this analysis.

12. Methods: Please ensure that the study is reported according to the STROBE guideline, and include the completed STROBE checklist as Supporting Information. Please add the following statement, or similar, to the Methods: "This study is reported as per the Strengthening the Reporting of Observational Studies in Epidemiology (STROBE) guideline (S1 Checklist)."

13. Methods: Did your study have a prospective protocol or analysis plan? Please state this (either way) early in the Methods section.

14. Results: Please provide both 95% CIs and p values for all results presented in the text.

15. Results: “Sleep duration ≥9 hours at age 60 (1.57, 1.17-2.09) and 70 (1.45, 1.05-1.98) years was associated with higher risk of multimorbidity” Please also present the result for sleep duration greater than 9 hours at age 50.

16. Results: “the pattern of associations was similar for analyses using sleep duration at age 50, 60, and 70 years although some of the associations were under-powered.” We suggest providing more detail on how the analyses were determined to be under powered, or changing the wording to reflect that while the results of these analyses were directionally similar, they were no longer statistically significant.

17. Results: “Accelerometer data and covariates were available on 3920 participants.” Please clarify the age group for this analysis. Please provide the HR, 95% CI and p values for this analysis.

18. Results: For example at: “Sleep duration ≥9 hours was also associated with a higher risk of multimorbidity (1.46, 1.07-1.99), although the association was attenuated after adjustment for behavioral and health-related factors (1.36, 1.00-1.86).” Throughout the results section, it is not always clear where the unadjusted results, or results from Model 1 or Model 2 are being presented. Please indicate which factors are adjusted for each analysis, unless this is consistent throughout.

19. Discussion: Please present and organize the Discussion as follows: a short, clear summary of the article's findings; what the study adds to existing research and where and why the results may differ from previous research; strengths and limitations of the study; implications and next steps for research, clinical practice, and/or public policy; one-paragraph conclusion.

20. Discussion: Here and throughout, please avoid any language that implies causality: “The analyses of health state transitions also indicated that long sleep at age 50 does not affect disease progression…”

21. Discussion: There was no evidence in the present data that short sleep duration plays an important role in disease prognosis…” Please clarify that you are referring to mortality.

22. Figures and Tables: Please be sure that there are titles and legends for each individual table and figure, including those in the Supporting Information. Please fully define any abbreviations used in figures and tables within the text of the legend.

23. Table 1 (and eTable 1 and eTable 2): Please report on the full set of characteristics of the study population (e.g. where there are more than two categories for each variable). Please provide a summary for the total group, in addition to the breakdown by sleep duration.

24. Table 2, Table 3, Table 4 and eTable 4, eTable 5, and eTable 6: Please report exact p values, instead of *p<0.05. Please also present the results from unadjusted analyses.

Comments from the reviewers:

Reviewer #1: This study examines sleep duration as a potential risk factor for first chronic disease, subsequent multimorbidity and mortality using data spanning 25 years.

Comments:

"Self-reported sleep duration was measured at age 50 and incidence of multimorbidity was defined as presence of two or more of 13 chronic diseases."

Can the authors please comment on the potential for self reporting bias?

"In 2012-2013 an accelerometer sub-study was undertaken on participants who attended the central London research clinic or were assessed at home if they resided in the South-Eastern regions of England. Wrist-worn accelerometers, the GENEActiv (Activinsights Ltd, Kimbolton, UK), were worn 24h over 9 consecutive days [22]. Sleep duration was estimated using a validated algorithm guided by a sleep log [23]; data from the first and last nights were removed leading to data over 7 nights. Usual daily sleep duration was estimated as the mean of sleep duration over the 7 nights and for those with less than 7 7 nights of measurement, weighted average of sleep duration was calculated as: 5 x week night sleep duration + 2 x weekend night sleep duration)/7. "

Are the authors able to explore and comment on how this accelerometer data compares to self-reported estimates of sleep duration?

"The association of sleep duration at age 50, 60, and 70 with risk of incident multimorbidity was examined using Cox proportional-hazards regression with age as the time scale in participants free from multimorbidity at the measurement of sleep duration. Data were censored at date of multimorbidity diagnosis, death to account for competing risk [30, 31], or 31st of March 2019, whichever came first. Age at the beginning of the follow-up was the age at clinical assessment closest to target age in the analyses (50, 60, and 70 years) from which the sleep duration measure and covariates were drawn. The proportional hazards assumption was verified using Schoenfeld residuals. Analyses were adjusted first for sociodemographic measures (Model 1), and then additionally for behavioral and health-related factors (Model 2)."

The authors have applied rigorous modelling methods, appropriately adjusting for potential confounding and reporting the results accurately. 

"To examine the robustness of our findings, we repeated the main analysis (1) in participants free from any chronic disease at the sleep measurement, (2) excluding users of sleep medication and (3) by investigating the association between accelerometer-assessed sleep duration at mean age 69 years and incident multimorbidity."

and

"In additional analyses, we examined the association of sleep duration after onset of a first chronic disease with transitions to multimorbidity and death, again using a multi-state model. The follow-up here started at the measure of first sleep duration following the onset of a first chronic disease. We repeated this analysis using inverse probability weighting to account for missing data that arise from the fact that 10 sleep duration was not measured after a first chronic disease and before onset of multimorbidity in some participants [34]"

The authors have suitably conducted additional analyses which help to demonstrate the robustness of the study findings.

"No association was found for mortality among those with existing chronic diseases. "

Did the authors consider exploring an overall transition analysis with death modelled as the outcome for the whole cohort (i.e. not conditional on the non-existence or existence of chronic disease(s))?

"In analyses excluding participants with any of the 13 chronic diseases at measurement of sleep duration (eTable 4), the pattern of associations was similar for analyses using sleep duration at age 50, 60, and 70 years although some of the associations were under-powered."

Can the authors please further expand on the power analysis they refer to here?

"The shape of the association between accelerometer-assessed sleep duration and incident multimorbidity confirmed that 7 hours of sleep was associated with the lowest risk of incident multimorbidity (Figure 3). "

Can the authors please comment on Figure 3b, which shows adjusted CIs crossing zero, suggesting no evidence of an increased HR for < or > 7 hours sleep?

Table 1 (and eTables 1 and 2): Can the authors consider additionally presenting BMI as a continuous variable here please?

Reviewer #2: 

This is a prospective cohort study addressing the role of sleep duration in the progression from a disease-free state to chronic disease, multimorbidity, and death shows short sleep duration to be a risk factor for the onset of chronic disease and multimorbidity but not for subsequent mortality in those with chronic disease(s). The paper is well written and with an extensive and detailed statistical analysis. I would like to point only one minor suggestion, please see the details bellow.

Page 9 - Statistical analysis, section Association between sleep duration at different ages and incident multimorbidity, last line "…with 7-hour sleep as the reference to examine the shape of the association between sleep duration and dementia risk."

This sentence does not seem to make sense here at this paper since assessing dementia risk was not an objective of this paper. Dementia was only assessed among the remain chronic diseases. 

Reviewer #3: This 25-year population-based cohort study leverages data on 13 chronic diseases to assess the relation between (self-reported) sleep duration and multimorbidity at different ages. Using sophisticated multistate models, they are able to show that short sleep duration (shorter or equal to 5 hours) increases the relative risk of any chronic disease and multimorbidity with 20% compared to longer sleep duration, but not to death.

The White Hall II study has several key strengths, including long-term, meticulously acquired health data on thousands of individuals. It also repeatedly assessed sleep behavior at various ages, and what I particularly like: a simultaneous assessment of objective sleep duration with that of self-report, allowing verification of self-reported data, as well as the authors' endeavor to repeat analyses with inverse probability weighting. Finally, I would like to congratulate the authors with their careful selection of diseases to define multimorbidity, while rightfully excluding risk factors.

I have three main concerns, and several minor suggestions to further improve the paper.

1) I like the analyses and idea of this study, but I struggle with interpreting the relevance of findings for public health or clinicians. Assessing relations with general health markers such as physical activity, sleep or diet with multimorbidity seems troublesome for etiological purposes since it is hard to say what specific disease (combinations) particularly drive associations. As this study appears to be of etiological origin (presentation of relative risks, and "Our findings support an etiological role of short sleep duration for onset of chronic diseases, including a first chronic disease and subsequent multimorbidity"), it seems that we would like to know how short sleep duration pathophysiologically relates to what specific diseases. This provides clinicians a direct target for intervention.

2) Only short sleep duration (equal or less than 5 hours) related to disease and multimorbidity. It raises the question why particularly this cutoff was chosen, as 5 hours is exceptionally short for an average weekday, and indeed prevalence is low (7.5%). It would rather be interesting to study the relationship continuously, by entering sleep duration per 1 hour into the models. 

3) The authors use sophisticated statistical models to disentangle relations over the life course. It is perhaps valuable to explain to the (general) reader why these models are relevant for this particular study compared to for instance traditional cox models. This can be done briefly in the introduction, and more elaborately in the methods part.

Abstract

-Rather the increased risk of multimorbidity associated with long sleep duration in those with existing disease might reflect the need for longer sleep in those with underlying chronic conditions

-This is an interesting interpretation, but it makes me wonder why this should not also hold for the relation with death?

-briefly add some demographics of the study population, including prevalence of short sleep duration as well as limitations of the study

Introduction

-although a number of outstanding questions remain regarding the nature of this relation. I would then rephrase "one, multiple.." to "First, multiple.." to increase readability.

Methods:

-Through linkage with medical records, data on a host of diseases are available. Were diagnoses also verified by study physicians? Is there any data available on the accuracy of these records?

-Apart from inverse probability weighting, how did the authors handle missing data in the main analyses?

-Perhaps add STROBE checklist

-add analysis with E-value, to address potential residual confounding that could not be adjusted for

Discussion

-Is there any data available on sleep quality? This seems relevant to include as authors specifically relate to "Long sleep duration might reflect poor overall sleep quality that could have a detrimental impact on health".

-Limitations of study are mentioned but could be expanded:

-if data on sleep quality is not present, this should be added

-perhaps the authors can reflect on the diversity of their study in terms of ethnicity?

-Since duration does not directly tell us something about quality, I am not sure whether I agree with the following statement: "Our findings suggest short sleep duration in midlife and old age is associated with higher risk of onset of a first disease, and subsequent multimorbidity among those with an existing condition. This supports promotion of good sleep hygiene in both primary and secondary prevention." Can the authors rephrase? What should clinicians do based on these findings, sole advice to sleep longer or stay longer in bed seems not so relevant if patients are not able to sleep.

Reviewer #4: The present study analyses the association between sleep duration and risk of first chronic disease, multimorbidity and death. While the analytical approach seems robust, there are several aspects that may threaten the internal validity of the findings.

- The main problem I see with this study is the low number of chronic diseases taken into consideration. This increases the risk of reverse causality given the potential heterogeneity in subclinical health states leading to and departing from a state of multimorbidity. If the authors cannot account for more diseases due to data availability issues, they should consider adjusting their models for some other comprehensive measure of health, e.g., walking speed, number of medications, muscle strength, etc. 

- One still wonders whether the potential impact of sleep duration on incident chronic disease and multimorbidity would still exist above and beyond the link with specific chronic conditions. One possible way to verify this is that the authors repeat the analyses after removing each of the 13 chronic conditions from their count, one at a time.

- Simply measuring sleep duration in absolute terms seems suboptimal. Data on sleep quality (e.g., sleep efficiency, sleep latency) have been shown to be important aspects of sleep. Moreover, assessing sleep duration in absolute terms could result in bias for those people with usual shorter/longer sleep duration. Also, what is the stability of self-reported sleep duration across time? Given that the authors have access to several of these measures over time, it may be pertinent to check this; even more so considering the long follow-up time for the 50-year and 60-year groups (22.6 and 13.4 years respectively). Last, what is the correlation between the subjective and objectively measured sleep duration? This should also be feasible to answer with their data and could shed further light into the validity of the findings.

- Chronic pain and depressive symptoms are well-known causes of sleeping problems and are, at the same time, closely linked to the development of multimorbidity. In order to untangle these potential confounding effects, could the authors repeat their analyses adjusting for these conditions?

Other minor comments:

- In the abstract, the authors mentioned that "Self-reported sleep duration was measured at age 50", but they later clarify this was measured at ages 50, 60 and 70 in the main text. Please update this information in the abstract.

- Why do authors call it age 50, 60 and 70 if 10-year approximations have been used? I would have rather worked with (and labelled them as) quinquagenarian, sexagenarian, septuagenarians subjects if the authors wish to make such an approximation.

- Revise the following sentence in the methods section concerning the dementia outcome: "we used restricted cubic spline regressions with Harrell knots [32], Stata command xblc [33], with 7-hour sleep as the reference to examine the shape of the association between sleep duration and dementia risk".

[LINK]

---

## [Decision Letter · Decision Letter 2]

9 Sep 2022

Dear Dr. Sabia,

Thank you very much for re-submitting your manuscript "Association of sleep duration at age 50, 60, and 70 with risk of multimorbidity: 25-year follow-up of the Whitehall II cohort study" (PMEDICINE-D-22-00972R2) for review by PLOS Medicine.

I have discussed the paper with my colleagues and the academic editor and it was also seen again by 3 reviewers. I am pleased to say that provided the remaining editorial and production issues are dealt with we are planning to accept the paper for publication in the journal.

[LINK]

We look forward to receiving the revised manuscript by Sep 16 2022 11:59PM.   

Sincerely,

Caitlin Moyer, Ph.D.

Associate Editor 

PLOS Medicine

plosmedicine.org

Requests from Editors:

1. Response to reviewers: Please address the remaining comment of reviewer 4.

2. Title: Thank you for revising the title. It may be helpful to also include the study setting in the title: Association of sleep duration at age 50, 60, and 70 with risk of multimorbidity among adults in England: 25-year follow-up of the Whitehall II cohort study”. Or similar.

3. Financial disclosure: Please include the URL of each funder website.

4. Data availability statement: Thank you for revising the data availability statement. If possible, please also include a contact email address where data access questions may be directed.

5. Abstract: Methods and Findings: Please ensure that all numbers presented in the abstract are present and identical to numbers presented in the main manuscript text.

6. Abstract: Methods and Findings: Please mention the important covariates adjusted for in the analyses (e.g. socio-demographic measures, behavioral and health-related factors).

7. Abstract: Methods and Findings: Please report p values to 2 decimal places for values of 0.01 or greater, and to 3 decimal places for values between 0.001 and 0.01.

8. Abstract: Line 49-50: Please consider mentioning some limitations of the study design, for example the potential for reverse causality, issues with generalizability, or self-reported sleep data could be mentioned here.

9. Abstract: Line 54-55: We suggest tempering this sentence slightly, given possibility of reverse causality: “Our findings suggest an association between short sleep duration and multimorbidity.”

10. Author summary: Line 80: We suggest: “There was no clear evidence for an association between long sleep duration at age 50 and risk of chronic disease.”

11. Lines 514-529: Please remove the Competing interest statement, Funding disclosure statement, and Data availability statement from the main text. Please make sure all information is completely and accurately entered into the manuscript submission system.

12. Figure 1: In the legend, please include some explanation for the various transitions (A-E).

Comments from Reviewers:

Reviewer #1: Many thanks to the authors for considering and responding to each comment in turn, undertaking additional analyses and amending the manuscript as required.

Reviewer #3: Dear authors, thank you for your extensive and clear replies. I have no further comments.

Best

Reviewer #4: The authors have adequately responded to my comments.

Regarding my first comment related to the operationalization of multimorbidity based on 13 chronic conditions, while I understand the justification provided by the authors, I still think they should add a limitation in relation to the risk of reverse causality given subclinical health states leading to and departing from a state of multimorbidity. In other words, short sleep duration and sleep disturbances may in fact represent the symptoms of already established disorders (e.g., depression, arthritis) that were still undiagnosed at baseline.

[LINK]

---

## [Editor Report · Decision Letter 3]

13 Sep 2022

Dear Dr Sabia, 

On behalf of my colleagues and the Academic Editor, Sanjay Basu, I am pleased to inform you that we have agreed to publish your manuscript "Association of sleep duration at age 50, 60, and 70 with risk of multimorbidity in the UK: 25-year follow-up of the Whitehall II cohort study" (PMEDICINE-D-22-00972R3) in PLOS Medicine.

Please also address the following editorial points:

-Title: Thank you for revising, please add “years” for the ages, and please update the title in both the main manuscript file and the manuscript submission system: “Association of sleep duration at age 50, 60,

and 70 years with risk of multimorbidity in the UK: 25-year follow-up of the Whitehall II cohort study”

-Line 89, and throughout: Please remove spaces within reference brackets [1,2]. Please check and revise this throughout the main text.

PRESS

Sincerely, 

Caitlin Moyer, Ph.D. 

Associate Editor 

PLOS Medicine